# Comparison between Conventional Treatment Processes and Advanced Oxidation Processes in Treating Slaughterhouse Wastewater: A Review

**Jen Xen Yeoh [1], Siti Nurul Ain Md. Jamil [1,2,*], Fadhil Syukri [3], Mitsuhiko Koyama [4] and Mohsen Nourouzi Mobarekeh [5]**

1  Chemistry Department, Faculty of Science, Universiti Putra Malaysia, Serdang 43400 UPM, Selangor, Malaysia
2  Centre of Foundation Studies for Agricultural Science, Universiti Putra Malaysia, Serdang 43400 UPM, Selangor, Malaysia
3  Aquaculture Genetics and Breeding, Aquaculture Department, Faculty of Agriculture, Universiti Putra Malaysia, Serdang 43400 UPM, Selangor, Malaysia
4  School of Environment and Society, Tokyo Institute of Technology, 2-12-1 Ookayama, Meguro-ku, Tokyo 152-8550, Japan
5  Department of Natural Resources & Environmental Sciences, Islamic Azad University, Isfahan (Khorasgan) Branch, Isfahan 81595-158, Iran
*  Correspondence: ctnurulain@upm.edu.my

**Abstract:** The blooming of the world's human population and the transition of the human diet into a more westernized, high-protein diet has accelerated the production of slaughterhouse wastewater (SWW) as the number of meat processing plants (MPP) has increased in the past few decades. Conventional treatment processes (CTP) used in treating SWW, such as anaerobic processes, membrane processes, and electrocoagulation, have significant limitations, such as low treatment efficiency, tendency to foul, and high energy consumption, respectively. While advanced oxidation processes (AOPs) appear promising in replacing the former, they lack economic feasibility when used as a single process. In this paper, the limitations and disadvantages of the CTPs used in treating SWW influents are evaluated. The idea of utilising AOPs as a "complementary" step rather than a single process is also discussed. The review paper further explores the variability of different AOPs, such as Fenton, Electro-Fenton, Sono-Fenton, etc., and their respective strengths and weaknesses in counteracting the limitations of CTPs. The idea of incorporating resource recovery into wastewater treatment is also discussed towards the end of the paper as a means of generating additional revenue for the industry players to compensate for the high operation and maintenance costs of SWW treatment. The integration of a new-generation treatment process such as AOP into CTP while being able to carry out resource recovery is a future hurdle that must be overcome by scientists in order to produce a versatile, powerful, sustainable, yet financially feasible and operationally pragmatic treatment system.

**Keywords:** slaughterhouse wastewater treatment; conventional treatment processes; advanced oxidation process; resource recovery

## 1. Introduction

Slaughterhouse wastewater, or SWW, is a type of wastewater produced by meat processing facilities (MPPs) from their slaughtering and facility-cleaning activities [1]. Barrera et al. reported that approximately 65% of the water used in slaughterhouses is associated with cleaning activities, whereas the remaining 35% is linked to maintaining personal cleanliness, keeping the water scald tank from overheating, sterilizing apparatus and other washing purposes [2]. According to the Environmental Protection Agency of the United States (US EPA), SWW is among one of the most detrimental wastewaters in existence [3]. SWW can have a wide range of compositions ranging from animal residues,

such as meat, feathers, fat, skin, and blood [4], to chemicals, such as pharmaceuticals, dyes, drugs, and some other sanitization products (e.g., detergent and disinfectants). Babu et al. reported dangerously high arsenic toxicity levels in SWW [5], whereas Avery et al. reported the persistence of the disease-causing Escherichia coli O157 among organic wastes, both of which could have severe impacts on human life [6].

If such wastewater is released into the receiving aquatic bodies without the necessary treatment having been carried out, it can severely cut the dissolved oxygen (DO) levels in the water, which can harm aquatic organisms [7]. Additionally, SWW also typically contains high concentrations of nutrients in the form of nitrogen (N) and phosphorus (P), which—when released untreated—could cause undesirable eutrophication and subsequently, an algal bloom [7]. Many have reported that SWW contains a high amount of soluble organics, as indicated by its elevated biochemical oxygen demand (BOD) and chemical oxygen demand (COD) [8,9], which could constitute a momentous threat to nature. Thus, they must undergo one or more rigorous, effective, and significant treatment processes before being deemed safe enough to be disposed of into any receiving aquatic bodies, such as lakes, rivers, or seas.

The blooming growth of the world's population and the increasing demand for a more westernized, protein-rich diet are the key factors in the increasing production of SWW. It has been reported that global production of meat products such as pork, beef and poultry has doubled in the past 10 years, and is projected to exhibit steady growth until the year 2050 [10]. The food and beverages (F&B) industry has long been known to be one of the largest consumers of water, and the meat industry consumes the most among them, requiring an astonishing 2422 $Gm^3$ of water per year globally. The beef cattle sector alone is already responsible for almost one third of the aforementioned figure [10,11]. To cater to such an enormous market, the number of MPPs must also grow to fulfil the supply requirements. This, in turn, creates a larger amount of SWW which requires treatment.

To combat the detrimental effects of the SWW, many countries around the world now impose progressively stricter regulations and limits for effluent emission, which caused the advancement of wastewater treatment technology to be more imperative than ever. Each country employs different types of combinations of processes in SWW treatment depending on regulatory limits, the best available technology (BAT), and the strength of the SWW [3].

SWW treatment and management systems increase the capital required to run a meat processing business and heighten the O&M cost in the long-run, which could result in a negative financial impact [12]. Abdelmoula et al. reported that using conventional activated sludge (CAS) to treat wastewater has incurred more than USD 0.8 million/year of operation cost to the industry players. Operation of membrane-based technologies, such as membrane bioreactors (MBR), easily set MPP owners back by more than USD 1.1 million/year due to the costs of periodic membrane chemical cleaning and replacement. Other costs, such as maintenance, materials, chemicals, and energy, also add to the final bill [13]. Additionally, Lyu et al. too mentioned in their work that the total cost for wastewater treatment alone in 2016 for the Hangzhou Bay Shangyu Economic and Technological Development Area (HSEDA), a typical chemical industrial park in China, was USD 97 million, accounting for almost 29% of the tax revenue of the park, for which energy, chemicals, and labour accounted for more than 60% of the total sum of treatment costs [14]. This shows that the usage of energy and chemicals incurs a large sum of fiscal deficit upon the relevant industries. Despite this, organic compound and nutrient recovery from SWW are relatively hidden and unresearched and could be part of the solution to compensate for the aforementioned painstakingly high operation and maintenance (O&M) costs of SWW treatment and management systems.

Several reviews have been previously conducted on SWW, e.g., Musa and Idrus compared and contrasted the strengths and weaknesses of several physical and biological processes commonly used in treating SWW [15]; Yaakob et al. investigated the characteristics and parameters of a particular slaughterhouse wastewater source as compared to a few well-cited sources [16]; and Baker et al. have highlighted the advantages of several

advanced technologies for the treatment of poultry SWW [17]. Comparison between the weaknesses of the conventional treatment processes (CTPs) that have long been employed in the industry and the strengths of different variants of new-generation advanced oxidation processes (AOPs) has not been performed before. This article aims to identify the limitations of CTPs used in treating SWW and the strengths of various AOPs compared to the CTPs mentioned. Lastly, the article reviews the current trends in resource recovery and the efforts that have been made to incorporate it into some treatment processes (conventional or advanced) as a means of revenue generation for a more economically feasible and sustainable wastewater treatment solution.

## 2. Parameters and Statistics

SWW is generally characterized by its high levels of proteins, fats, carbohydrates, and nutrients derived from the meat, blood, feathers, or skin of the animal, often accompanied by detergents, sanitisers, and antibiotic compounds. The bulk parameters used in defining SWW strength may include (but are not limited to) total nitrogen (TN), biochemical oxygen demand (BOD, typically $BOD_5$), total phosphorus (TP), chemical oxygen demand (COD), total suspended solid (TSS), total organic carbon (TOC), and other physical parameters, such as colour, foulness, turbidity, etc. [11,18,19].

The ranges of the characteristics of SWW, as sourced from three different journals, are tabulated below in Table 1.

**Table 1.** Typical parameters used in determining SWW strength and their range and mean values.

| Parameter | Source 1 (Typical) [1] | | Source 2 (Actual, Location: Parit Raja, Malaysia) [16] | | Source 3 (Actual, Location: Jelutong, Malaysia) [20] | |
|---|---|---|---|---|---|---|
| | Range | Mean | Range | Mean | Range | Mean |
| BOD (mg/L) | 150 to 8500 | 3000 | 1341 to 1821 | 1602 | 573 to 1177 | 875 |
| COD (mg/L) | 500 to 16,000 | 5000 | 3154 to 7719 | 5423 | 777 to 1825 | 1301 |
| TOC (mg/L) | 50 to 1750 | 850 | 195 to 652 | 419 | NR | NR |
| TN (mg/L) | 50 to 850 | 450 | 163 to 564 | 361 | 154.6 to 362.4 | 258.5 |
| TP (mg/L) | 25 to 200 | 50 | NR | NR | NR | NR |
| TSS (mg/L) | 0.1 to 10,000 | 3000 | 378 to 5462 | 3438 | 395 to 783 | 589 |
| K (mg/L) | 0.01 to 100 | 50 | NR | NR | NR | NR |
| Color (mg/L Pt Scale) | 175 to 400 | 300 | NR | NR | NR | NR |
| Turbidity (FAU) | 200 to 300 | 275 | NR | NR | NR | NR |
| pH | 4.8 to 8.1 | 6.5 | 7.3–8.6 | 8.02 | 6.3 to 6.9 | 6.6 |

FAU = formazine attenuation unit, NR = not reported.

As tabulated in Table 1, it can be seen that the mean values of the various parameters of the two raw wastewater samples collected from two different sites in Malaysia (Source 2 and Source 3) are well within the range of the typical values as reported by Bustillo-Lecompte and Mehrvar, 2017 (Source 1). While the values shown here are typical, some SWW can exhibit extremely high strengths, e.g., the cattle SWW samples used by Musa et al., which exert $32,000 \pm 112$ mg/L of COD, $17,158 \pm 95$ mg/L of $BOD_5$, $22,300 \pm 212$ of TSS, and $915 \pm 18$ of TN. This shows that SWW can have widely varying degrees of pollution depending on the area, type, and capacity of the MPP from which the influent was sampled. Regardless, low-strength SWW is still considered a safety hazard due to the detrimental impact it can have on aquatic bodies despite its low strength.

To counter the environmental impact caused by MPPs, instructions and ordinances are of utmost importance. Baker et al. have compared and contrasted the limitations and

standards set by many agencies for wastewater effluent discharge, including Malaysia, as presented below (Table 2).

**Table 2.** Limitations and standards set by several world agencies [17].

| | Parameter | | | |
|---|---|---|---|---|
| | BOD$_5$ (mg/L) | COD (mg/L) | TSS (mg/L) | TN (mg/L) |
| WB Standards | 30.00 | 125.00 | 50.00 | 10.00 |
| EU Standards | 25.00 | 125.00 | 35.00 | 10.00 |
| US Standards | 26.00 | NR | 30.00 | 8.00 |
| CA Standards | 5.00–30.00 * | NR | 5.00–30.00 * | 1.00 |
| AU Standards | 6.00–10.00 | 3 * BOD5 | 10.00–15.00 | 0.10–10.00 |
| MY Standards | 20.00 | 120.00 | 50.00 | NR |

* The Canadian standards for BOD$_5$ and TSS in the to-be-discharged effluents are 5, 20, and 30 mg/L in freshwater lakes and slow-flowing streams; rivers, streams, and estuaries; and shorelines, respectively. WB: World Bank; EU: European Union; US: United States of America; CA: Canadian; AU: Australian; MY: Malaysia; NR: not reported.

From the table, it is observed that disparate countries have dissimilar regulations and limits which range according to the classification of food, agricultural, and industrial wastewater [6]. Of these, the standards set by the World Bank are the lowest, whereas the Australian Standards, as set by the Australian Environmental Council, are among the most stringent.

## 3. Conventional Treatment Processes Used in Treating SWW and Their Inherent Limitations

Conventional treatment of SWW is not much different compared to the popular technologies used in metropolitan wastewater and may include preliminary, primary, secondary, and even tertiary treatment. The basal step in SWW management is to minimize the number of inputs [21]. Locating and minimising the production of SWW at its source before any treatment takes place is usually recommended.

### 3.1. Preliminary Treatment

In preliminary treatment, a majority of large, solid objects are removed from the wastewater. Typical apparatuses used in preliminary treatment may include sieves, screeners, and strainers. Thus, sizable solids with a diameter ranging from 10 to 30 mm are withheld, while the SWW are allowed to pass through. This alone can reduce the BOD by up to 30% [18]. After preliminary treatment, the SWW can be used directly for land application or be sent to subsequent primary and secondary treatment.

### 3.2. Land Application

The land application technique involves the direct application of SWW into agricultural lands [18]. This will provide the crops with necessary nutrients. However, a major drawback is that this method is affected by temperature and location [21]. For instance, for countries that are located in temperate regions, this method may not be feasible during the cold season. As such, the SWW must be stockpiled throughout that time, and this will incur additional costs due to storage and transportation. Other disadvantages may include a change in landscape aesthetics; the presence of foul odour, defiled soil, and potential surface and groundwater pollution; and the cultivation of pathogenic microorganisms [6].

### 3.3. Primary Treatment Methods
Physicochemical Treatment

The primary treatment method used is the physicochemical treatment of SWW. Physicochemical treatment encompasses the segregation of the SWW into several parts, archetypally the detachment of solid particles from the liquid. One of the most commonly used

methods for the primary treatment of SWW is the dissolved air floatation (DAF) method. The method separates solids from the mixture by gassing air from the bottom of the SWW influent. Lighter lipid-based compounds and solid particles are carried to the top of the tank, where they form a sludge layer which is subsequently removed as scum via the scrapping technique [17]. This is extremely efficient at reducing the levels of lipid-based compounds, TSS, and BOD in SWW [1,21]. Li et al. reported the elimination of 25.8% COD and 31.6% BOD from the starting loads of wastewater by DAF alone. However, this is restricted by several major drawbacks, such as high electricity consumption and the utilisation of one or many aeration devices. Ratnayaka et al. reported that floatation procedures consume 0.050–0.075 kWh of energy per cubic metre of water treated [22] compared to anaerobic processes, which utilise virtually no energy. The use of chemicals (e.g., $H_2O_2$ as a source of gas) in the setup also indirectly makes the sludge unusable. Additionally, this method is frequently faces system failures and inefficacious solid–liquid separation, which makes it simply unfeasible due to the accompanying high cost of operation and maintenance [15].

Other methods may include sedimentation or coagulation–flocculation (CF), electrocoagulation (EC) and membrane techniques such as reverse osmosis (RO), microfiltration (MF), and ultrafiltration (UF). [21]. However, these all have weaknesses as well. Sedimentation and coagulation–flocculation employ a large number of chemicals (coagulants and flocculants) and generate huge amounts of sludge, which can be difficult to handle and requires additional processes to treat. Electrocoagulation calls for high energy demand, and as such, displays low cost-efficiency, while membrane technologies are frequently plagued with membrane fouling problems which require chemicals to clean and hence may bring about secondary pollution [15] and financial deficit.

### 3.4. Secondary Treatment Methods
Biological Treatment

Biological treatment "treats" wastewater by relying on decomposers to degrade the organic wastes into simpler substrates. Biological treatment can be segregated into aerobic processes and anaerobic processes. Aerobic systems remove organic materials in condition in which oxygen, $O_2$, is abundant. They are relatively popular as they operate at high rates of reaction. Some common aerobic processes include aerobic sludge blanket reactors (SBR) and activated sludge (AS). On the other hand, anaerobic systems rely on microorganisms to break down organic waste in the absence of oxygen into methane gas ($CH_4$) and water. Anaerobic systems are sometimes chosen over their aerobic counterparts because of their lower cost due to no sophisticated aeration device being necessary. They also produce less sludge and provide excellent $BOD_5$ eradication with potential for the recovery of biogas [23]. However, the efficiency of anaerobic processes is highly limited by the strength of SWW. While biological processes can reduce TN and TP levels, extra post-treatment is often needed to reduce the TP and TN levels to an acceptable standard [24,25]. Biological processes are also prone to toxin inhibition as the microbes responsible for metabolizing the organic pollutants can be easily rendered inactive by the presence of toxic compounds such as benzene, toluene, dichloromethane, etc. [2] Some of the commonly used anaerobic processes are anaerobic filtration chambers (AFC), anaerobic lagoons (AL), anaerobic sequencing batch reactors (ANSBR), anaerobic baffled reactors (ABF), upflow anaerobic sludge blanket reactors (UASBR), etc. Figure 1 shows the flow chart on the treatment of SWW, starting with minimising the input.

Table 3 below shows a summary of the inherent limitations of conventional treatment processes used in treating SWW:

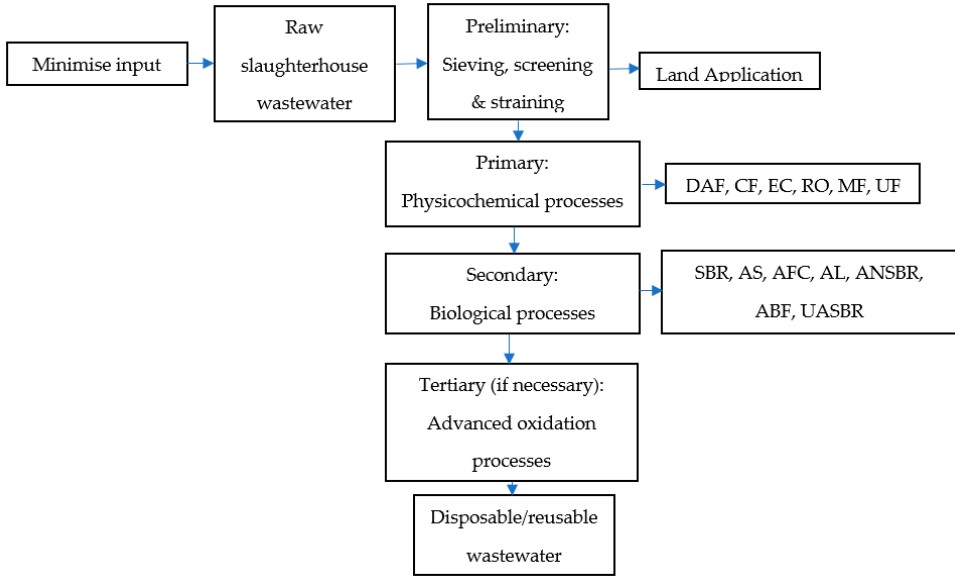

**Figure 1.** Flow chart on the treatment of SWW, starting with minimising the input.

**Table 3.** The inherent limitations of the conventional treatment processes used in treating SWW.

| Category | Process | Limitations |
|---|---|---|
| Physicochemical | Dissolved Air Floatation | - Consumes a large amount of electricity<br>- The use of chemicals renders sludge unusable<br>- Frequent breakdowns<br>- Freezing issues |
| | Coagulation-Flocculation | - Uses a large quantity of coagulating or flocculating chemicals<br>- Generates a huge volume of sludge, which requires additional processes to treat<br>- Sludge can only be disposed of by land application or burning means |
| | Electro-coagulation | - Consumes a large amount of electricity<br>- Meagre cost efficiency |
| | Membrane technologies | - Perpetual biofouling problem<br>- Requires chemicals to clean, causing secondary pollution<br>- Low energy efficiency in treating SWW |
| Biological | Aerobic | - High start-up costs due to aeration devices<br>- Consumes a large amount of electricity<br>- Generates a huge volume of sludge, which requires additional processes to treat<br>- Sludge can only be disposed of by land application or burning means<br>- Prone to toxin inhibition<br>- An unbalanced nutrient ratio (COD: N: P) may impede the feasibility |
| | Anaerobic | - Highly dependent on weather, geography, and accessibility to large spaces<br>- Long hydraulic retention time (HRT)<br>- Long start-up period<br>- Low efficiency<br>- Effluent does not satisfy the discharge limit (N and P) and may require post-treatment<br>- Odour problems<br>- Prone to toxin inhibition |

## 4. Advanced Oxidation Processes (AOPs), Their Variants, Strengths, and Weaknesses

Recent research widely reported that conventional treatment methods, such as biological methods and physicochemical methods, have weaknesses. For example, disinfecting SWW may require chlorination, which may lead to the formation of toxic byproducts, which cause secondary pollution [26]. Del Nery et al. and the UN Division of Sustainable Development claimed that the effluents treated by conventional treatment processes have unsatisfactory TN and TP levels and require a second advanced treatment for safer disposal [27]. On the other hand, Musa et al. observed that the efficiency of conventional treatment systems is crippled at higher organic loading rates, characterized by slow-growing microorganisms. This, in turn, results in a lengthy start-up period, and the formation of scum and sludge washout [8]. Lastly, Brillas et al. stated in their work that failure in removing these recalcitrant compounds, which they named "persistent organic pollutants (POPs)" in wastewater treatment plants with typical physicochemical and biological processes, reinforced by their high resistance to UV and surrounding temperature, causes them to accumulate in oceans, rivers, lakes, and drinking water, usually at the level of $\mu g \, L^{-1}$ to $ng \, L^{-1}$ [23]. This statement is further supported by Barrera et al., who claimed that "failure in removing these recalcitrant compounds" may lead to incompliance with effluent discharge limits and water reuse standards [2], which may bring about legal issues. This led scientists to turn toward a new generation of treatment processes called advanced oxidation processes.

Advanced oxidation processes, or AOPs, are water and wastewater treatment methods that harness the oxidation potential of in-situ-produced hydroxyl ($\bullet OH$) or sulphate ($\bullet SO_4^-$) radicals to remove organic contaminants existing in the aqueous medium. $\bullet OH$-based AOPs are the most popular and most widely accepted due to their remarkable oxidation potential ($E°(\bullet OH/H_2O) = +2.80$ V), which is just below the oxidation potential of fluorine (+2.87 V) [23,28,29]. Sulphate-radical-based AOPs have also gained popularity due to their effectiveness in degrading the majority of organic pollutants. This is because sulphate radicals have a much longer lifespan of 30 to 40 µs compared to that of hydroxyl radicals, which is reported to be a mere 20 ns [30].

AOPs have been garnering the public's attention to serve as a replacement for conventional treatment methods or as a complementary post-treatment method to current biological processes. AOPs can inherently achieve disinfection without the need of adding any chemical water disinfection. This can prevent the emergence of toxic derivations [31]. As such, AOPs have been given recognition as sophisticated mineralising, water-recycling, and pollution-controlling agents capable of achieving remarkable overall results as a complementary method to biological processes [31,32]. For instance, Alfonso-Muniozguren et al. suggested that electrochemical advanced oxidation processes (EAOPs) may very well be the ultimatum for SWW treatment due to their capability in removing recalcitrant compounds [33]. Babu et al. reported that advanced oxidation processes (AOPs) are regarded as efficacious for the mineralisation of recalcitrant compounds by empowering their biodegradability and diminishing their toxicity [5]. Brillas et al. also stated in their work that AOPs are considered a "green" method since organics are obliterated via mineralization upon the assault of in-situ-produced hydroxyl radicals. The high standard redox potential of this radical guarantees that it is capable of mineralising most organic wastes [23] and nutrients. All of these are pieces of evidence showing that AOPs are one of the most promising new-generation treatment methods and that they have a boundless potential to replace conventional treatment methods.

To the best of the authors' apprehension, there are very few reports in the literature in which AOPs are used alone in treating SWW. Instead, many employ AOP as a complementary "polishing" step or incorporate advanced oxidation combined processes (AOCPs) in their SWW treatment system. Alfonso-Muniozguren et al. used an activated sludge–filtration–ozonation (A-F-O) system to treat pretreated slaughterhouse wastewater. They have mentioned in their work that ozone, through oxidation, can eliminate micropollutants as well as microorganisms without tampering with the nontoxic nature of the treated

effluents (e.g., by adding chemicals) and is also capable of being virucidal. This can be achieved by using ozone alone or combined with other AOPs. In COD and BOD, 93% and 98% reductions were achieved, respectively, obtaining final values of 128 mg/L COD and 12 mg/L BOD. TSS was successfully reduced by up to 99% using the same system, giving the final value of just 3 mg/L. Interestingly, a 98% reduction in phosphorus (P) and a full inactivation of total coliforms (TC) was recorded after just 17 min of ozonation [11,34,35]. AOPs have many times been reported to be used as a complementary step to reduce TN and TP levels. Gray et al. used $TiO_2$/UV photocatalytic AOP to complement conventional ultrafiltration (UF) process and obtained 90–97 % TP removal efficiencies for municipal wastewater [36]. Bustillo-Lecompte et al. used an anaerobic baffled reactor–aerobic activated sludge combined process complemented by $UV/H_2O_2$ (ABR-AS-UV/$H_2O_2$) on a sample of real slaughterhouse wastewater and achieved a 91.29% of maximum TOC removal and 86.05% of TN removal [37]. In another interesting study by Besharati Fard et al., the degradation of slaughterhouse wastewater using an upflow anaerobic sludge blanket (UASB) reactor with a hydraulic retention time of 26 h produced a total chemical oxygen demand (TCOD) and phosphate removal efficiencies of 62.2% and 36.5%, respectively. However, when complemented by Fenton as a post-treatment method under optimum conditions, the combined method eliminated TCOD and phosphate levels up to 98.6% and 90.5%, respectively. It is worth noting that while the Fenton process alone is already capable of removing TCOD and phosphate up to 95.41 and 85.29%, respectively, it is rarely applied as a single process; it is usually applied as a complementary step due to economic considerations. Nonetheless, the study demonstrated that AOP is extremely capable of reducing organics and nutrients, either as an individual process or as a complementary step [38].

In another work by Alfonso-Muniozguren et al., it was reported that it is essential to advance a "novel, clean and efficient" technology that allows the appropriate treatment of SWW since it cannot be fully treated by using conventional physicochemical processes. As such, they explored the feasibility of electrochemical oxidation (EO) and EO-related processes, either alone or combined, as a polishing step for SWW. SWWs which had been previously subjected to grit removal, fat removal, biological treatment, and settling—but failed to comply with the EU discharge standards for treated metropolitan wastewater concerning organics, suspended solids (SS), and colour—are subjected to different variants of EO processes. Some of these include EO with hydrogen peroxide (EO/$H_2O_2$), EO with ultraviolet C light (EO/UVC), and EO with ultraviolet C light and hydrogen peroxide (EO/UVC/$H_2O_2$). Without preozonation, it took more than 480 min for EO, ~400 min for EO/$H_2O_2$, ~260 min for EO/UVC and ~120 min for EO/UVC/$H_2O_2$. COD and SS were below the European emission limit values as of the treatment time. All of the above examples show that AOPs are excellent complementary steps to basic physicochemical or biological treatment methods and that the more sophisticated the methods are, the less time it requires to treat a certain amount of SWW sample [33].

Figure 2 shows some common AOP variants used in wastewater treatment.

Table 4 shows a compilation of recent articles (2016–2021) that describe different variants of AOP that have been used for water or wastewater treatment followed by a simple description and its corresponding reaction details and equations.

The strengths and weaknesses of the 10 different variants AOPs aforementioned are also summarised in Table 5.

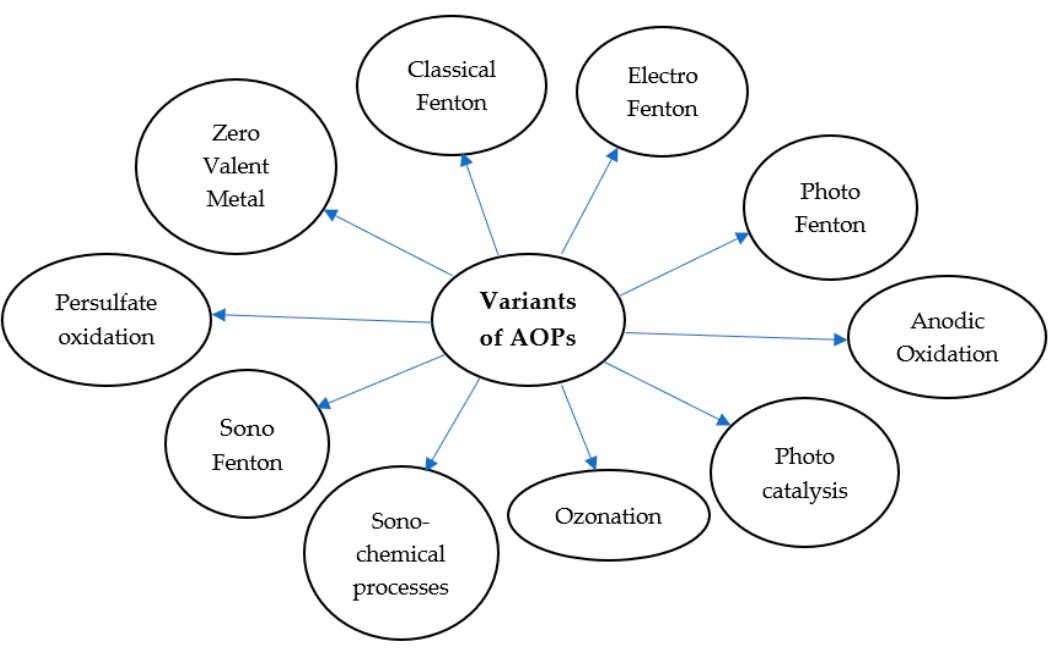

**Figure 2.** Some common AOP variants.

**Table 4.** Different variants of AOP and their descriptions.

| No. | AOP Variant | Description | Reaction Details and Equations | Relevant Articles Published in the Recent 5 Years (2016–2021) |
|---|---|---|---|---|
| 1 | Classical Fenton | One of the oldest AOP processes. This process generates hydroxyl radicals from the reaction between iron (II) ions and hydrogen peroxide at pH = 3. The efficiency of this process is highly affected by pH. | Fenton Process $Fe^{2+} + H_2O_2 \rightarrow Fe^{3+} + OH^- + \bullet OH$ Regeneration of iron (ii) ions $Fe^{3+} + H_2O_2 \rightarrow Fe^{2+} + H^+ + \bullet HO_2$ | [39–43] |
| 2 | Electro-Fenton | A variant of Classical Fenton where $H_2O_2$ is generated in situ in the electrolyte by supplying oxygen at the surface of the cathode under an acidic medium. | In-situ generation of hydrogen peroxide: $O_2 + 2\,H^+ + 2e^- \rightarrow H_2O_2$ Regeneration of iron (ii) ion by the cathodic reduction of iron (iii) ion: $Fe^{3+} + e^- \rightarrow Fe^{2+}$ | [42,44–47] |
| 3 | Photo-Fenton | An improved version of Classical Fenton in which hydroxyl radicals are produced from both the Fenton process as well as from the degradation of hydrogen peroxide in the presence of UV light (photolysis). | Photolysis: $H_2O_2 + h\nu \rightarrow 2\,\bullet OH$ ($\lambda < 300$ nm) Regeneration of iron (ii) ion: $Fe(OH)^{2+} + h\nu \rightarrow Fe^{2+} + \bullet OH$ ($\lambda < 450$ nm) | [47–52] |
| 4 | Anodic Oxidation | Hydroxyl radicals are generated by the oxidation of water in the presence of high-$O_2$-evolution overvoltage anodes | Water oxidation at the anode surface: $M + H_2O \rightarrow M\,(\bullet OH) + H^+ + e^-$ M: Anode | [53–55] |

**Table 4.** *Cont.*

| No. | AOP Variant | Description | Reaction Details and Equations | Relevant Articles Published in the Recent 5 Years (2016–2021) |
|---|---|---|---|---|
| 5 | Photo-catalysis | Generation of hydroxyl radicals and other reactive oxygenated species by shining UV light over catalysts such as $TiO_2$, $ZnO$, and $ZnS$. | Photoexcitation: $Cat + h\nu \rightarrow Cat\,(e^- + h^+)$ Production of hydroxyl radicals from external oxidants ($H_2O_2$) $H_2O_2 + Cat\,(e^-) \rightarrow OH^- + \bullet OH$ | [56–62] |
| 6 | Ozonation and catalytic ozonation | Organic waste can be eliminated either via the direct attack of ozone or via the hydroxyl radicals generated under alkaline conditions, which promotes the decomposition of ozone. Catalytic ozonation incorporates a catalyst which allows the decomposition of ozone even at lower pH. | Catalytic Ozonation: $Fe^{2+} + O_3 + H_2O \rightarrow Fe^{3+} + OH^- + \bullet OH + O^2$ | [63–67] |
| 7 | Sonochemical/ Ultrasound processes | Ultrasonic irradiation leads to the cavitation phenomena, which is the formation, growth, and subsequent aggressive collapse of microbubbles or cavities, generating extremely high temperatures and pressures in the process. The violent collapse of the cavities then promotes the formation of reactive hydroxyl radicals via the dissociation of the water molecule. | Thermal dissociation of $H_2O$ in the presence of ultrasound: $H_2O\,+))) \rightarrow \bullet OH + H\bullet$ | [68–71] |
| 8 | Sono-Fenton | An improved version of Classical Fenton in which hydroxyl radicals are produced from both the Fenton process as well as from the cavitation process in the presence of ultrasound (sonolysis). | Fenton Process $Fe^{2+} + H_2O_2 \rightarrow Fe^{3+} + OH^- + \bullet OH$ Regeneration of iron (ii) ions $Fe^{3+} + H_2O_2 \rightarrow Fe^{2+} + H^+ + \bullet HO_2$ Thermal dissociation of $H_2O$ in the presence of ultrasound: $H2O\,+))) \rightarrow \bullet OH + H\bullet$ | [72–74] |
| 9 | Persulfate/ Peroxymono-sulfate oxidation | Production of reactive sulphate radicals via the decomposition of persulfates or peroxymonosulfates. This process can be accelerated by catalysts such as heavy metal, UV, ultrasound, or heat. | Persulfate activation by iron $S_2O_8^{2-} + Fe^{2+} \rightarrow Fe^{3+} + SO_4^{2-} + \bullet SO_4^-$ | [62,75,76] |
| 10 | Zero valent metal (ZVM)/$H^+$/$O_2$ | Under the acidic condition, zero-valent metals, such as iron and aluminium, undergo corrosion and generate hydrogen peroxide, which then further decomposes in the presence of zero-valent metal to generate hydroxyl radicals. | Corrosion of zero valent metal: $2\,Al^0 + 3\,O_2 + 6\,H^+ \rightarrow 2\,Al^{3+} + 3\,H_2O_2$ Decomposition of hydrogen peroxide in the presence of ZVM: $Al^0 + 3\,H_2O_2 \rightarrow Al^{3+} + 3\,OH^- + 3\,\bullet OH$ | [77–82] |

**Table 5.** Strengths and weaknesses of the different variants of AOP.

| AOP Variants | Strengths | Weaknesses |
|---|---|---|
| Classical-Fenton | - Iron and $H_2O_2$ are relatively safe when optimally dosed<br>- Simple operating principles<br>- Does not require mass transfer | - Highly sensitive to pH, must be carried out at pH = 3.<br>- Efficiency decreases sharply when pH increases as Fe precipitate out as $Fe(OH)_3$<br>- The dosage of $H_2O_2$ must be optimised<br>- Sludge problems |
| Electro-Fenton | - In situ production of $H_2O_2$ eliminates the need to store, transfer, and handle<br>- Permits the studies on the degradation mechanism by controlling the kinetics<br>- Minimised sludge production<br>- Economically feasible | - Generation of $H_2O_2$ is slow<br>- Current efficiency is reduced at a pH lower than 3<br>- $Fe^{2+}$ regeneration is lethargic |
| Photo-Fenton | - Increased hydroxyl radical production rate<br>- Increased rate of degradation of pollutants | - The use of an artificial source of light can hurt the financial aspects<br>- Sensitive to pH<br>- The intensity and wavelength of UV radiation can significantly impede the degradation rate |
| Anodic Oxidation | - Capable of treating large volumes of wastewater<br>- No pH limitations<br>- High degradation efficiency | - Aromatic solutions are barely degraded, as hardly oxidizable carboxylic acids are generated<br>- Costly<br>- Electrode fouling |
| Photocatalysis | - Direct inactivation of disease-causing bacteria<br>- High treatment efficiency<br>- Low-cost catalysts (e.g., TiO2) | - Full-scale application of photocatalysis is scarce due to the difficulties in catalyst recovery<br>- Some catalysts are highly toxic and may cause secondary pollution<br>- Quantum yield is relatively low due to the fast recombination of electron–hole pairs |
| Ozonation and catalytic ozonation | - Direct inactivation of diseasing-causing bacteria<br>- High treatment efficiency | - Full-scale application of catalytic ozonation is scarce due to the difficulties in catalyst recovery<br>- Catalytic ozonation mechanisms are not well understood |
| Sonochemical/ Ultrasound processes | - Avoidance of electrode fouling<br>- Excellent mass transfer capacity | - Highly energy intensive<br>- Low energy efficiency<br>- Hard to be carried out on a large scale<br>- Noise problems |
| Sono-Fenton | - Swift degradation of refractory organic compounds<br>- Removal of free hydroxyl radicals | - Low mineralization efficiency<br>- The emergence of intermediate byproducts<br>- Difficult to set up<br>- Noise problems |
| Persulfate/ Peroxymonosulfate Oxidation | - Relatively longer lifespan of the persulfate radicals<br>- Can work under mild pH conditions (4–9)<br>- Comparable redox potential to OH radicals | - Production of toxic byproducts<br>- High level of persulfate ions<br>- Complex quenching reactions |
| Zero valent metal (ZVM)/$H^+$/$O_2$ | - Large surface area and surface reactivity<br>- Versatile applications<br>- Nanoscale ZVM is particularly promising | - Risks of toxic effects |

## 5. Incorporating Resource Recovery into Wastewater Treatment

Bustillo-Lecompte and Mehvrar suggested in their work that the meat processing industries (MPPs) must assimilate elements of both waste minimisation and resource (i.e., biogas or nutrient) recovery into their SWW management and treatment strategies considering the enormous portion of the industrial wastes and byproducts which may have a potential for direct reuse, including nutrients as fertilizers and/or methane gas as biofuel [1]. Recently, resource recovery in the form of biofuels or nutrients has garnered considerable attention due to the increased costs of the environmental and financial aspects of energy as well as the availability and cost of both mineral and synthetic fertilizers [83]. The focus has been slowly shifting from "wastewater treatment" alone to "wastewater treatment and resource recovery". This section discusses the available technologies or methodologies associated with resource recovery, with insights into the problems faced in implementing them.

Cai et al., in 2012, utilised microalgae to recover nutrients from wastewater streams. Microalgae have long proven to be extremely effective and efficient at removing pollutants, such as nitrogen, phosphorus, and even toxic metals, from a wide variety of wastewater. Cai et al. reported that extensive studies of algae growth have been conducted in municipal, agricultural as well as industrial wastewaters [84–86]. Compared to other biofuel feedstocks, microalgae have the following advantages: (1) there exists no competition between the microalgae and the crops for cultivable land and freshwater as they can also be cultivated in saline water and on barren land; (2) microalgae have both high growth rates and high lipid contents of 20–50% on a dry weight basis [87]; (3) microalgae, being photosynthetic, are capable of carbon dioxide fixation and thus have the added benefits of reducing greenhouse gases and improving the quality of air; (4) microalgae, with their high growth rates, can quickly use up the nutrients from most wastewaters, providing a complimentary method for wastewater treatment, particularly in nutrient removal; and (5) upon lipid extraction, microalgae cultivation generates a byproduct called algae biomass residue. It can be utilized as a source for nitrogen, e.g., as a protein-rich animal fodder or as a fertilizer for vegetations [88]. In summary, microalgae cultivation is extremely versatile, with applications ranging from biofuel generation, $CO_2$ fixation, and mitigation to wastewater treatment [89].

On the other hand, Brennan et al., just recently in 2020, conducted a review on the recovery of viable ammonia–nitrogen ($NH_3$–N) products from agricultural SWW by membrane contactors. Brennan et al. mentioned in their work that nitrogen can be recovered via several methods, namely ion exchange, microwave radiation, air stripping, and using hydrophobic membrane contactors [90]. Membrane contactors are given special emphasis in Brennan's work, in which they stated that they are capable of using a stripping solution to capture the nitrogen in the form of $NH_3$, which can be used to produce ammonium ($NH_4^+$) salts afterwards. Brennan also mentioned that studies have shown that $NH_4^+$-based salts can be regenerated to make liquid fertiliser, which would benefit the farmers in promoting their crops' growth [91,92]. Brennan et al. aim to fill the research gap left behind by the works of previous scientists who have been working on SWW, such as Bustillo-Lecompte and Mehvrar [1] and Mittal et al. [18], by reconciling nitrogen removal and membrane technology by way of a critical comparison to conventional methods.

Nitrogen recovery is much more intricate since there is no accessible precipitate [93]. However, the use of membranes makes up for this, as studies have shown that it is capable of recovering nitrogen up to 99% in the form of $NH_3$ while rejecting other contaminants. Hence, a feasible fertilizer byproduct can be obtained [94]. There are several advantages associated with the usage of membranes, as stated in Brennan's work. Membranes are capable of achieving removal efficiencies of 85.8, 50, 97.5, and 99.8% for COD, BOD, total phosphorous, and total nitrogen, respectively, of SWW. On the other hand, their low cost brings about a reduction in the total cost of the SWW treatment system while partially producing economically feasible byproducts, such as ammonium salt fertilizers, which could generate extra revenue. Credit has also been given to membrane technologies for

their simple operation, excellent selectivity, and low energy consumption [92]. Additionally, membrane contactors inherently have a driving force to keep the transfer ongoing due to the huge partial pressure differences between the feed side and the permeate side [92,95]. The hydrophobic nature of the membrane also restricts the permeation of liquid water while allowing vapour to pass through from the feed side of higher partial pressure to the permeate side of lower partial pressure. However, despite the advantages, Brennan et al. did mention in their work that membrane contactors, like all other membrane-based technologies, are prone to customary problems, i.e., high initial setup cost, potential pump failure, and the infamous membrane fouling; the last of these is particularly problematic given how susceptible they are to frequent fouling and therefore wetting, which may ultimately impede efficiency [92,96,97].

Hülsen et al. (2013), on the other hand, utilized enriched purple phototrophic bacteria (PPB) or purple non-sulphur bacteria (PNSB) to remove N ($NH_4^+$), P ($PO_4^{3-}$) and COD with the aid of infrared (IR) radiation under anaerobic conditions. This method successfully achieved an impressive 99.6% reduction in N, whereas P and COD removals were at slightly lower levels, standing at 88% and 63%, respectively. While conventional COD removal methods release organic carbon in the form of $CO_2$, PPB can assimilate the carbon removed into usable biomass instead of oxidizing it into $CO_2$ gas. $NH_4$-N and $PO_4$-P were also assimilated into the biomass rather than simply being destructively oxidized into other compounds. Hülsen et al. furthered their research by adding acetate as an additional carbon source to test the maximal N and P elimination potential, which did not display any signs of compromise [83].

Sreyvich et al. examined the efficiency of N and P recovery via struvite precipitation. Struvite precipitation is known for reducing nutrient pollution in soil and water but at the same time producing slow-releasing and high-quality fertilizers [98]. Additionally, Martí-Herrero et al. used low-technology tubular digesters to produce biogas from SWW treatment. Martí-Herrero et al. mentioned in their work that most of the research reported in the literature is feasible at a laboratory scale but often not realizable at a real-life large scale. Additionally, in scientific journals, mesophilic and thermophilic conditions are usually used, which may be not plausible when the amount of sample is too large as heating a large amount of sample would require an enormous amount of energy. Martí-Herrero et al. suggested the usage of low-technology digestors, also known as low-cost digestors, in real-life applications. They often lack sophisticated mixing devices and/or heating systems, which makes their usual working environment psychrophilic. Typical low-cost digestors may include anaerobic lagoons (AL) as these have a relatively low initial cost, negligible O&M cost, and are simple to operate. The analysis of the full-scale, low-cost tubular digestors with biofilm carriers shows that the digestor is competitive in biogas production in anaerobic digestion of SWW when the organic loading rate (OLR) is less than 0.5 kg $COD/m^3d$ and is competitive for % COD removal when the OLR is less than 0.25 kg $COD/m^3d$. While the final COD value of the SWW effluent after a 75% reduction at a hydraulic retention time (HRT) of 20 h still stands at 900 mg/L and requires post-treatment, the results certainly showed that it is possible to develop a rather "mediocre" technology which sacrifices some efficiency but gives better accessibility for the underprivileged community [99].

Aslam et al. examined the feasibility of anaerobic membrane bioreactors (AnMBRs) for urban wastewater treatment. In their study, a wide range of contaminants can be removed for the reuse of water, while the degradation of organics via an anaerobic route can yield $CH_4$-rich biogas for energy production. However, optimisations and improvisations are still direly required as the process still faces six major issues, as highlighted in the article—namely, (1) organic strength of wastewater, (2) fouling issues, (3) salinity accumulations, (4) inhibitory materials, (5) temperature, and (6) membrane durability. [100]

In the authors' opinion, while the current treatment technologies are considered successful (with COD, $BOD_5$, TSS, TN, and TP removal efficiencies $\geq$ 90%) in treating SWW, the incorporation of resource recovery into treatment processes is still relatively unresearched and contains hurdles to be overcome. Additionally, resource recovery is predominantly available

and incorporable in biological methods (anaerobic processes) only. In that sense, the combined process of biological processes and a sophisticated treatment method—e.g., AOPs—may actually yield a stronger synergy than the individual processes. In the aforementioned publication by Bustillo-Lecompte, they used an anaerobic baffled reactor–aerobic activated sludge combined process complemented by $UV/H_2O_2$ (ABR-AS-$UV/H_2O_2$) on a sample of real slaughterhouse wastewater and achieved 91.29% of maximum TOC removal and 86.05% of TN removal. While the removal efficiencies are already excellent by themselves, the combined process also yielded an additional 55.72% of maximum $CH_4$ biogas recovery [37]. Indeed, the future research perspectives lie well within the conciliation of an advanced treatment process and resource recovery for an economically more feasible and sustainable solution that neither individual AOPs nor individual biological processes can achieve. The article by Qadir et al. summarises how the projected increasing wastewater production (24% by 2030 and 51% by 2050 over the current threshold) demands imminent attention to develop a well-formulated resource recovery plan with a maximum return on investment (ROI) to offset global water, nutrient, and energy needs [101].

## 6. Conclusions & Recommendations

This review provides the reader with an introductory knowledge of SWW production and the conventional treatment processes (CTP) available to treat it. Various limitations of CTPs are also highlighted, thereby allowing the readers to acknowledge their weaknesses at a glance. Special emphasis has been given to different variants of AOPs, with their working principles, strengths, and weaknesses evaluated, thereby enabling future researchers to explore wider research questions for future work.

In conclusion, up-to-date advanced oxidation processes can be considered extremely versatile SWW treatment methods that incorporate many benefits in one package, e.g., strong oxidising (thus "cleaning") power, low cost and ease of use, relative environmental friendliness, no requirement of the addition of external chemicals, and many sub-AOPs to choose from. However, individual AOPs still have limitations, so this is where combined AOP processes shine, as combined AOPs can generate an arsenal of different reactive species that could lead to complete mineralization and more thorough removal of trace pollutants with possible resource recovery, such as AOP–biological combined processes.

Truly, the future of AOP technologies requires increasingly crucial progress in the resource recovery sector be assimilated into the SWW treatment system as well as advancement in the current AOP technologies to yield cleaner, stronger, more cost-efficient, and more easily employable sub-AOP processes which may prove to be more reasonably employable methods for MPP owners. As such, while future research perspectives should be focused on developing better AOPs, the importance of the incorporation of resource recovery into the treatment system should not be left aside. Indeed, the conciliation of these two is a future research question that requires answering. However, the selection of the resource recovery technology still depends on several factors, such as the best available technology (BAT), business capital, geographical limitations, and the stringency of legislative regulations.

**Funding:** This research and its article processing charges (APC) were funded by Universiti Putra Malaysia (UPM) under the research grant 'Geran Putra Berimpak' with UPM code number GP-GPB/2021/9699200 with reference number UPM.RMC.800-3/3/1/GP-GPB/2021/9699200, Vote No. 9699200.

**Data Availability Statement:** The authors confirm that the data supporting the findings of this study are available within the article.

**Acknowledgments:** Thanks are due to the Chemistry Department, Faculty of Science, UPM; Centre of Foundation Studies for Agricultural Science, UPM; Aquaculture Genetics & Breeding, Aquaculture Department, Faculty of Agriculture, UPM; School of Environment and Society, Tokyo Institute of Technology and Environmental Science Department, Islamic Azad University, Isfahan (Khorasgan) Branch.

**Conflicts of Interest:** The authors declare no conflict of interest.

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
