# Peer review of "Comparison between Conventional Treatment Processes and Advanced Oxidation Processes in Treating Slaughterhouse Wastewater: A Review"

_water, doi:10.3390/w14223778_

Round 1
Reviewer 1 Report
Please see the attachment.

Author Response
Author's notes
- The introduction section has been updated. However, if the reviewer still thinks that it could still use a little more update, it would be very kind to highlight as in which section or which line of the manuscript the reviewer thinks require updating.
- Some opinions were added, wherever applicable. Kindly see lines 85-98, 428-447
- Summary maps were added, kindly see figures 1&2.
- The authors went through the articles suggested and found that the articles deal with heavy metal removals. Since this manuscript focuses on SWW, heavy metal removals are not commonly mentioned in SWW. The authors apologise for not being able to include them in this manuscript.
- The recommendation of adding a glossary of abbreviations is duly noted. However, to futureproof, kindly allow the addition to be implemented once and once if the manuscript has been confirmed for acceptance.
- Grammar has been checked and corrected, wherever applicable.
Reviewer 2 Report
In this manuscript, the authors reviewed a series of studies on the comparison between conventional treatment processes and advanced oxidation processes in treating slaughterhouse wastewater. review on advantage and disadvantageous are reviewed. The presented manuscript misses the quantitative comparison between AOP and conventional methods. The manuscript can be improved with writing skills and the flow which is missing in the manuscript. The significance of the work is not impactful for the publication. The review doesn't add up to the knowledge base in this field. To my opinion, after evaluation, I feel the novelty and scientific content of this manuscript are still not enough for publication in this Journal.
Author Response
Thank you for the review
Reviewer 3 Report
There is a major issue with the topics covered in the manuscript. First, the "conventional" treatment methods are discussed but apparently only considers the organic matter removal in the scope of biological treatment. In 2022 reducing the biological process to organic matter removal is quite obsolete. Both nitrogen and phosphorous can be removed via biological means. Enhanced biological phosphorous removal was discussed first in 1960s, thus it it younger than the activated sludge method from the 1910s but still a mature technology. On the other hand, the paper does not address the unbalanced nutrient ratio (COD:N:P) which limits the feasibility of the so called "conventional" activated sludge method. Also, if one wants to go by the traditional terminology, biological treatment is a secondary method and physical and physico-chemical methods are primary.
Second, the AOP section focuses on organic matter removal and within that the persistent organic pollutants without discussing the effect of mentioned AOPs on nitrogen and phosphorous. There is only one instance where phosphorous (orthophosphate) removal is mentioned despite this sentence in the AOP section "... the effluents treated by conventional treatment processes should undergo a second advanced treatment to reduce TN and TP concentrations in the SWW for safer disposal (Baker et al., 2020; Del Nery et al., 2016)." The authors should look into the N and P treatment ability of the researched AOPs. The title of the paper suggests that both conventional treatment and advanced oxidation processes were tested on slaughterhouse wastewater in the papers reviewed by the authors but the most of them on AOP dealt with different types of water, thus slightly misleading the reader.
Thirdly, section 5 discusses resource recovery but it is in no connection with the title or the previous two sections. Clearly, instead of end-of-pipe approach, recovery should be in the focus for a feasible and sustainable wastewater treatment solution neither the conventional solutions (if sludge is disposed and not used for energy or as fertiliser) the AOP methods (because they mechanism is mineralisation) are not suited for this task. This is barely addressed in the text and it is not shown, how any of the compared technologies can be combined or replaced by resource recovery regarding slaughterhouse wastewater.
Other issues with the manuscript:
Reference style - the authors use secondary sources numerous times, listing both the review and the reviewed papers as references. This is a bit confusing for the first read, and not a common solution. Furthermore, instead of clearly indicating which source belongs to which notion (in case of enumerations), the authors put all references at the end of the sentence (e.g. lines 346-348).
Gendered use of reference mentions: instead of Babu and his coworkers, Avery and her team, the authors should use a neutral and more academically accepted approach, Babu et al. (2019), Avery et al. (2005).
Water uses the ACS style, with number in order of appearance, in square brackets. If the authors applied that for the first version of manuscript, it would have been apparent that several papers were cited numerous times (Baker et al., 2020 and Babu et al., 2019 were found 7-7 times in the text). While it is unavoidable in some instances, the frequency occurring in this manuscript reduces the quality of a review paper substantially.
The expression "power-hungry" - concrete values should be indicated to allow comparison instead of using such a vague expression. Same goes goes for high operation and maintanence costs - how much is painstakingly high?
Information in Table 3 is not backed up sufficiently, especially the statements on resource recovery and power use. If DAF sludge is not suitable for recovery why is that not a limitation for AOP?
There is an indication in the text that the AOPs could be used on their own for the treatment of slaughterhouse wastewater, but the data in the paper do not support this, on the contrary. This is not discussed in the paper later.
To sum it up, the manuscript is more of a description of literature of treatment methods that could come into question for slaughterhouse wastewater and not a comprehensive review with critical insight on the topic. It lacks to provide novelty, therefore the manuscript cannot be accepted in its current form.
Author Response
First and foremost, the authors would like to express gratitude for such a well-scrutinized and meticulously written review report. The authors took the review seriously and have made appropriate improvements.
- The authors did not overlook the capabilities of biological processes to remove nitrogen and phosphorus, on top of organics. As seen in now line 202, "While biological processes CAN reduce the TN & TP levels...". However, what the authors are trying to suggest is that, as much as biological processes can reduce nitrogen and phosphorus levels, they are much inferior to AOPs. Indeed, in now lines 283-291, it can be seen that an up-flow anaerobic sludge blanket (UASB) reactor gives 36.5% phosphate removal efficiency while Fenton under optimum conditions can provide 85.29%. The combined process of UASB-Fenton pushes the value to 90.5%, showing that AOPs are far superior to biological processes in means of removing nutrients, whether as a single process or as a complimentary process. 'Unbalanced nutrient ratio' has been addressed in table 3 as one of the limitations of conventional aerobic processes. Biological processes are also now categorised under 'secondary methods".
- The effects of AOPs on nutrient removals have been addressed. Kindly see lines 263-289. Regarding the issue of the slow diversion from SWW to a broader "different types of water". This is actually one of the advantages of AOPs compared to CTPs. As AOPs rely on the generation of reactive radicals to mineralise pollutant species and the mode of attack is rather unselective. This goes to suggest that while AOP is highly suitable for the treatment of SWW, it is also suitable for various other kinds of wastewater, due to its non-selective nature, e.g., textile wastewater, municipal wastewater & pharmaceutical wastewater.
- The issue where section 5 is seemingly disconnected from the title and the previous sections is addressed. As the current special issue is titled “...advanced Oxidation/Reduction and Biological Processes...”, emphasis must be given to the comparison of biological processes (as one of the CTPs) to AOPs, hence the word "resource recovery" did not appear in the title albeit its appearance in the manuscript in section 5. Nevertheless, the authors have also amended section 5 to make it seems more connectable. The section now starts with the importance of incorporating resource recovery into wastewater treatments as means to generate revenue for feasibility and sustainability (as suggested), then moves into a review of the efforts that have been made in resource recovery narrowed down to the author's opinion on how the conciliation of resource recovery to current treatment processes remains a big hurdle although biological-AOP combined processes appear promising as it satisfies the stringent treatment efficiencies yet is able to leverage on methane gas recovery.
- The referencing style has been amended, wherever applicable.
- The gendered use of reference mentions has been fully removed.
- The manuscript now uses ACS referencing style, as employed by Water. The occurrences of frequently-cited journals (such as Babu et al. and Baker et al. ) have been cut down.
- Actual values have been added. Kindly see lines 75-80, and 173-175.
- Indeed, resource recovery remains a challenge to be incorporated into AOP. However, as mentioned in 3., the advancement of combined processes may have a strong synergistic effect that empowers the advantages of individual processes while diminishing their weaknesses. This is addressed in lines 428-447, where the authors mentioned that this is future research perspectives --- to make use of combined processes instead of individual ones.
- The authors believe that while AOPs are fully capable to be used on their own for the treatment of SWW, it is not economically feasible due to the relatively high cost of AOP. As such, the authors propose that while AOP is capable of as a single process, it is BEST used as a complementary process. This is mentioned several times in the manuscript. Kindly see lines 263 onwards.
The authors again thank the reviewer for such constructive comments. Please let us know what you think about the revised edition.
Reviewer 4 Report
The presented work focused on the comparison between conventional treatment processes and advanced oxidation processes in treating slaughterhouse wastewater.
I would like to point out that the presented topic is very interesting and is a significant contribution to the field.
However, some the manuscript should be improved before the publication. Please, see the comments below:
1. Although it is a review paper, the aim of the work should be presented.
2. The novelty of the work performed was not emphasized. How is this work different from previous review articles on this topic?
3. Undoubtedly, the quality of the Tables should be improved.
4. It is a review paper, hence, the number os cited work is not sufficient. Indeed, the Authors did not cite many works published in the last few years. For instance:
DOI: 10.1111/1477-8947.12187
- The issue of wastewater treatment should be better discussed. Please see the following recently published papers:
- Regarding the use of membrane techniques for treatment of wastewaters, please see the following recently published papers:
DOI: 10.3390/en15144981
DOI: 10.1016/j.scitotenv.2021.149612
DOI: 10.3390/membranes12030275
5. In conclusion, research perspectives should be discussed.
6. Please, try to present some graphs.
Author Response
Author's Notes
- The aim of the work is now addressed in lines 93-98.
- The novelty of the work is now addressed in 85-93
- The tables have been improved. The authors apologize for the previous dreadful table qualities (compatibility issues).
- Some of the suggested articles have been added to the manuscript. Kindly see lines 421-427 and 443-447.
- The research perspectives are slightly touched on in lines 428-447, and also mentioned in the conclusion section
- As much as the authors would want to present graphs, as requested. The authors did not find appreciable content to add. However, figures were added as summary maps (figures 1 & 2).
Round 2
Reviewer 1 Report
After the first run of revisions, the manuscript improved in quality. However, I will recommend it to be published until the following comments are addressed.
1. Compared with the previous Review, the author should clearly explain the improvement of this work.
2. Reduce the number of tables and use Figs to present data as much as possible.
Author Response
During the first revision, the manuscript was reviewed by 4 experts in the field. So, many changes have been made after the first revision. While it is not possible to address all of them. The authors try to be meticulous as possible in explaining them.
- The citation style has been changed to ACS style as per Water's requirement.
- Citations from secondary sources such as reviews have been amended. The occurrence of frequently cited papers has been reduced.
- The research gap between this manuscript and the previous manuscript has been addressed (lines 94-107)
- The tables have been improved.
- Biological processes are now classified as 'secondary' processes
- Figures have been added to improve reading interests.
- Concrete data on the high power consumption of DAF has been added. (Lines 182-188)
- Lines 272 - 304 are added to clarify that while AOPs are superior even when they are used alone, they often are used as a complementary process (or combined process) due to economic concerns.
- Chapter 5 has been reworked to make it fit the title and the scope of the manuscript better.
- The author's opinions are added (as per recommended by other reviewers) in lines 437-456
As for the reduction of table issues, as the data presented are comparative data, the authors humbly think that it is best to be presented in table form. However, some figures have to be added to the present data, wherever applicable. The authors regret not being able to address this issue this time.
Reviewer 2 Report
Thank you for the revised version of your manuscript. I see that the authors have made some improvements in the manuscript, and I believe that the quality of the manuscript has improved after the revision. Advanced oxidation processes based on non-thermal plasma using various reactor designs, including falling film reactors are recommended to be compared and discussed. Hybrid advanced oxidation processes are also applied to enhance the refractory organic removal efficiency such as in (https://doi.org/10.1016/j.chemosphere.2019.04.160 ). Overall, now the revised manuscript is well-written and structured. The background of the introduction is sufficiently described. I suggest accepting the paper after the recommended amendment.
Author Response
The authors would like to express their utmost gratitude for the recommendation for publication. Thank you for the time and effort given in reviewing this manuscript.
Reviewer 3 Report
The authors took the comments of the previous review into consideration and substantially changed the manuscript. There are still a few reservations considering the approach of the authors to the topic.
For instance, the authors mention how much wastewater treatment costs the meat processing industry in the introduction. While factually true, this will not be solved by introducing AOP to replace or complement the commercially available technologies, only if circularity becomes a major design aspect. This is contradicted in the abstract: "Whilst advanced oxidation processes (AOPs) appear promising in replacing the former." Please revisit the abstract to reflect the article itself as the focus of the new manuscript is slightly shifted.
The authors bring up the need of chemicals and high energy consumption as disadvantages of the conventional treatment methods but does not give concrete values on the costs. Philipp, M., Masmoudi Jabri, K., Wellmann, J., Akrout, H., Bousselmi, L., & Geißen, S. U. (2021). Slaughterhouse Wastewater Treatment: A Review on Recycling and Reuse Possibilities. Water, 13(22), 3175. provide detailed information on several of the treatment technologies this manuscript covers and from that it is apparent that activated sludge can be in the same range as the AOP (UV/H2O2) albeit one of the cited references mention higher values.
In the reviewer's opinion, because the cost factor of the treatment is emphasised the original intention is distorted by it. AOP can overcome certain problems of the conventional methods but will not reduce the cost of treatment. The focus of the paper should be more on how and where AOP can assist the future of SWW treatment instead of the current phrasing from which sometimes seems to imply that AOP is superior to the other methods (which in some cases could be true but biological processes have progressed to be able to treat the SWW to the required effluent water quality in a cost-effective manner).
Author Response
Dear reviewer,
The authors would like to again thank you for your constructive comment. The following changes have been made, as per the recommendation.
- The abstract has been revisited and revised to fit the slightly shifted focus of the manuscript better.
- Concrete data on the high cost of energy consumption and chemical uses is addressed in lines 76-93.
- The manuscript actually does discusses some superiority of AOP compared to CTPs (cost-effectiveness aside) in lines 253 onwards. E.g., AOP can achieve disinfection without chlorination, as opposed to AS which utilises microbes and may bring about a health hazard albeit that not being the focus of the manuscript. Other disadvantages of CTPs such as AS such as long HRT, sludge disposal issues and lower efficiency are also addressed in table 3 under 'Biological processes.
Reviewer 4 Report
The manuscript has been corrected, hence, I recommend it for publication in present form.
Author Response

(The authors gave the same response as above.)

Round 3
Reviewer 1 Report
Accept in present form
Reviewer 2 Report
The paper has been revised well. it can be considered for publication